# Reducing Life Cycle Embodied Energy of Residential Buildings: Importance of Building and Material Service Life

**Abdul Rauf**

Architectural Engineering Department, United Arab Emirates University, Al Ain 15551, United Arab Emirates; a.rauf@uaeu.ac.ae

**Abstract:** Energy use in the building sector is considered among major contributors of greenhouse gas emissions and related environmental impacts. While striving to reduce the energy consumption from this sector, it is important to avoid burden shifting from one building life cycle stage to another; thus, this requires a good understanding of the energy consumption across the building life cycle. The literature shows greater emphasis on operational energy reduction but less on embodied energy, although both have a clear impact on the building's footprint and associated environmental impact. In previous studies the importance these energy aspects have been presented; however, the critical role of embodied energy linked to the replacement of materials over a building's life is not well documented. Therefore, there is a knowledge gap in the available the literature about the ways to reduce the embodied energy requirements of buildings over their useful life. Service life of buildings and their constituent materials may play an important role in this regard. However, their potential role in this respect have not been explored in the previous research. This study critically addresses the above-mentioned gaps in the literature by investigating the combined effect of building and material service life on life cycle embodied energy requirements of residential buildings. Life cycle embodied energy of a case study house for an assessment period of 150 years was calculated based on minimum, average and maximum material service life values for the building service life of 50, 100 and 150 years. A comprehensive input–output hybrid analysis based on the bill of quantities was used for the embodied energy assessment of the initial and recurrent embodied energy calculation for each scenario. The combined effect of building and material service life variations was shown to result in a reduction in the life cycle embodied energy demand in the order of up to 61%. This provides quantifiable and verifiable data that shows the importance of building and material service life considerations in designing, constructing, and managing the buildings in efforts to reduce energy consumption by buildings. A secondary contribution of this paper is a detailed sensitivity analysis which was carried out by varying the material service life values of each building material and the embodied energies for each new scenario was recalculated for two assessment periods. The findings show that, for each material service life variation, the LCEE increases as BSL increases for a 50-year assessment period, but the LCEE decreases for a 150-year assessment period.

**Keywords:** life cycle embodied energy; initial embodied energy; recurrent embodied energy; building service life; material service life; maintenance; sensitivity analysis

## 1. Introduction

The current global scenario as it applies to energy use, climate, and environmental pollution is both enormous and incrementally concerning due to far-reaching impacts on both plant and animal life [1–3]. In fact, a lot of negative environmental issues are attributed to energy consumption [4,5]. These trends characterize the situation in many countries of the world over the last decade. Thus, various researchers have sought to understand and mitigate the status quo; from the United States [6], China [7] Mexico [8], India [9] and Nigeria [10]. The common thread in these studies has been reviewed and the reports

suggest that active measures can be taken to simultaneously understand the impact of predisposing factors and promote development of practical solutions [11–13].

Increasingly, attention has been given to the role of buildings and its related activities from construction to operation and international reports show that buildings have significant impact on the environment [14–16]. According to the International Energy Agency (IEA) as at 2017, buildings were responsible for 36% of global energy use and 40% of energy-related carbon dioxide ($CO^2$) emissions [17]. The challenging scenario is exacerbated by uncertainties regarding the best way to address climate-related problems [18], combined with the rapidly fluctuating global energy landscape [19].

There is, however, another complication to this trend: by 2050, human pollution is expected to reach 9.7 billion [20]. Increase in population implies increase in housing, which further increases the considerable amount of energy already consumed in residential buildings. Quite fundamentally, there no doubt that a close relationship exists between population growth, energy consumption and urbanization [21,22]. It has been reported that energy consumption patterns are the direct result of household occupants' actions [23], although on a broader scale, this may also be influenced by external factors such as high energy subsidies [24]. Our focus here is to simply to situate the current study on the background of the human-environmental scenario, to show the strong interconnection between choice and impact.

Some reports suggest that in the near future, new housing developments will cover 230 billion square meters [17]. Without definitive action from the various actors in the building industry, a solution to this scenario, or a mitigation of the current trend is nearly impossible. Therefore, there is a need for strategic approaches which ensure the reduction of life cycle building energy consumption at all stages if the objective of saving the planet by reducing degradation and greenhouse gas emissions is to be achieved. In this regard, one approach reported in several studies is to explore and reduce operational energy consumed in buildings [25,26]. Due to the abundance of related studies, the accuracy of assessment measures that reduce operational energy have become very efficient. Additionally, policies have been drafted by several countries. In Denmark for example, building operational energy has been reduced by over 60% since the late eighties [27]. In the Gulf, countries such as the United Arab Emirates and Saudi Arabia have set ambitious clean energy targets [28,29] These all connect and comply with international agreements on environmental sustainability and accountability, supported by the United Nations conventions [30,31].

Nonetheless, the extensive effort towards curtailing the negative impact of buildings by focusing on operational energy has sidelined another critical component of building environmental impact. Comparatively speaking, based on the foregoing, the number of studies which have investigated the embodied energy of buildings or sought to develop new strategies for reducing its impact, is much less [32]. In some regions, the absence of data makes the investigation of this topic complicated, and thus often avoided [33,34]. Studies which have indeed investigated this aspect, report that its impact is substantial and argue that there should be extensive research in this area [35,36]. One added consideration is the use of sensitivity or uncertainty analysis in embodied energy the literature. Some studies have reported the benefits for understudying future variations in life cycle study parameters [37,38], or its applicability in conducting comparative lifecycle assessments [39], as well as combined operational and embodied energy assessments [37]. This contributes to the literature by elaborating the applicability of sensitivity analysis in a case study building, focusing on variations in the building, as well as material service life of the selected case study building.

When broken down, embodied energy is made up several components which are later discussed in this paper. However, a preliminary review of studies around this topic shows that only few give attention to "recurrent embodied energy" [35,40–42]. This component comprises of the embodied energy which is associated with building maintenance; it specifically covers the replacement or repair of building materials and components during the life span of a building. Thus, the life span of both building materials, and the building

itself, impact the recurrent embodied energy. Yet, few studies have investigated these factors sufficiently. Even the studies, which have addressed both these factors; they have separately analyzed building service life [42] and material service life [43]. This study provides a clear understanding of these aspects and their combined effect on life cycle energy demand which is not well investigated in the literature. The current study aims to fill this gap. The focus is to investigate the life cycle energy demand of residential buildings as a consequence of its service life, as well as the service life of its construction materials.

### 1.1. Life Cycle Embodied Energy Assessment

There are multiple studies which have been conducted to show the importance of a life cycle assessment from the generally viewing the building design and typology [44–46] and a few which focus on the impact of material selection [47–49].

In practice and academic the literature, the nomenclature used in to define the systematic procedure for quantitatively approximating the embodied energy of a building is known as the "life cycle embodied energy assessment" (LCEEA) [50–53]. Life cycle assessment can be divided into several phases. Typically, these stages are goal and scope definition, inventory analysis, impact assessment and interpretation [54]. In the "goal and scope definition" stage, purpose of the life cycle assessment is established. The "Inventory analysis" involves the collection of data regarding the environmental impact of the product. In impact assessment phase, the significance and magnitude of possible environmental effects based on the results of the "inventory analysis" stage are assessed. Finally, conclusions are drawn from the results of the inventory analysis and impact assessment.

This analysis is calculated over the life cycle of the building; it facilitates an understanding of different aspects of energy consumption during the useful life of the building and is closely linked with life cycle energy (LCE) assessments [43]. Due to the extensive nature of the LCEA, it can be adopted as a strategic approach to guide design modification and improvements, as well as the proper selection of materials and building assemblies. These steps can be useful in reduction of both the amount of energy consumed and also, greenhouse gas emissions linked with the building. This assessment protocol has been widely applied in the literature to investigate associated energy consumed in for various building typologies -residential, commercial, and educational [55–59].

The LCEEA is based on a defined system boundary to assess the energy consumed in a building, and comprehensively covers the manufacture of materials and products, construction and maintenance of the building, as well as the process of demolition at the end-of-life stage [60]. In general, the assessment covers three key aspects: the initial embodied energy at construction, the energy embodied due to subsequent material or component serving and replacement during the use, and finally, the energy consumed in demolition of materials or disposal of the same[61].

Approaches and Examples of Embodied Energy Assessment

In the case of buildings, significant amount of energy is consumed in various stages and multiple processes, when combined, these account for far-reaching impacts on the environment. In order to quantify both direct and indirect energy associated with the construction and the materials, respectively, a few methods are commonly used in the literature. These assessment methods are also used to calculate the embodied energy associated with the maintenance of the building after construction, and the material or component repairs or replacement in the building until it is demolished or disposed of.

The common methods include the process analysis, input–output analysis, and hybrid analysis [36,62]. In each of these, there are differences which relate to the system boundary and energy inputs, which form a major part of the assessment procedure [63]. These differences led to variations in the results of the assessment based on the method chose; thus, selecting the right method is critical. Previous studies have elaborated on the distinction between these methods [32,36,42]. In general, the process analysis is considered the most accurate while the input–output analysis is considered the most comprehensive. The

hybrid analysis is a combination of strengths from both approaches into a single assessment approach. Table 1 shows a comparison of these common assessment methods. It shows the strengths and weaknesses of each method based on sources in the literature.

**Table 1.** Comparison between the process and input–output analysis.

| Method | Strength | Weakness | Source |
|---|---|---|---|
| Process Analysis | Combines process, product, and location-specific data Classified as the most accurate | Details of major production data is sometimes unavailable Complex procedure due to upstream supply chain associated with the approach 59% truncation error reported | [36,45,64–66] |
| Input–output analysis | Energy is quantified based on economic data of product production processes Considered to be systemically complete and very comprehensive | Complicated by aggregation of dissimilar products Applicability is limited to a single product Prone to errors due to the use of economic data Multiple-time counting of energy embodied associated with delivered fuels | [36,62,63,65–67] |

There have been several studies in recent years which have focused on embodied energy assessment of buildings. Some of these studies are based on real case studies [65,68] while others are based on extended parametric simulations [69,70]. In some cases, the studies have applied one of the common methods above, while some studies applied the use of computer software in the assessment [68,71,72]. Findings of some of these examples are presented below with emphasis on embodied energy results.

A study by Azzouz et al. (2017) was carried out on an UK office building to develop and guide quantifiable early-design strategies which reduce life cycle energy, as well as carbon intensity in buildings [71]. The gross floor area (GFA) of the building was 11,550 m$^2$ with an assumed service life of 60 years. The researchers used the IMPACT software for the assessment and found out that embodied energy was 9% of the life cycle energy while operational energy was also calculated and accounted for 91% of the life cycle energy. The study also found out that embodied energy savings are highest when recycled building materials are used in the construction.

Another study by Lolli et al. (2017) investigated a 160 m$^2$ residential building in Norway with the objective to outline how a dynamic approach using a parametric analysis tool (PAT) could be adopted for extensive investigations on life cycle energy [73]. The study thus, reported that operational energy consumed was about 49% of the LCE, and embodied energy was about 51%. The authors used the Process Analysis for the assessment and reported that the developed PAT can facilitate the definition and design of optimum building envelopes. They also reported that the PAT is useful in comparatively conducting pre-assessment studies on multiple design solutions.

Furthermore, Wang et al. (2018) studied office buildings in Hong Kong; focusing on ten high-rise buildings, the study was carried out to investigate life cycle energy through a combined systematic the literature review and case studies analysis [68]. Using a hybrid life cycle analysis approach and the SimaPro software, the study found that the embodied energy ranges from 11 to 22%, while the operational energy ranges from 78 to 89%. Reviewing the data collected, the authors reported that high-rise buildings have twice as much embodied energy as low-rise buildings. They also reported that in terms of high energy intensity materials the top three are concrete, reinforcing rebar and finally structure steel.

The literature on embodied energy assessment figures differs for several studies and is influenced by multiple factors. Some of these factors include the type of assessment method [36,42], building material specification [39,45], as well as construction or structural material [39,45,74]. Others include the energy source [39], passive design strategy [75], house layout [45] and construction method [72].

In the current study, the focus is on the role played by building, and material service life on the embodied energy associated with residential building construction and maintenance. Thus, the next two sections provide a description and definition of these two concepts.

### 1.2. Building Service Life

As a way of definition "Building service life" refers to the time—usually measured in years, during which a constructed building is operated or occupied [76]. During this time, these building require to be serviced from time to time, including the maintenance and replacement activities of different building materials and assemblies. As the service life of the building increases, a corresponding rise in the number of types/replacement cycles related to servicing, replacing or maintenance of materials, components or systems invariably increases. This results in a rise in the recurrent embodied energy of the building. Similarly, as the building service life reduces, new or re-construction of the building is needed due to the occupancy requirements. This results in an increase of the initial embodied energy associated with the original construction over a specific assessment period [36].

A careful consideration of the estimated building service life at design stage can help to select the appropriate materials. Selection of appropriate materials at this stage allows the optimization of initial embodied energy. Due to the effect of building service life on recurrent embodied energy demand, building service life considerations can also help to decide the type and frequency of activities required to maintain, repair and replace building materials and systems. This approach is critical for optimizing the recurrent and life cycle embodied energy demand of buildings.

Considering the foregoing, the importance of building service life considerations cannot be overemphasized, particularly at the design stage since it would leave a lasting impact on the building's life cycle embodied energy as it relates to building servicing and maintainability [32].

In practical terms, the various assumptions guide the approximation of a building's lifespan which may also have a ripple effect of the proper material selection. For example, when a building structure is intended to provide a long service life, such as institutional buildings, it is expected that the durability of the building materials will be significantly high. However, temporary structures or structures in areas which may be repossessed or redeveloped soon, may logically be designed for a comparatively shorter lifespan. In these cases, it would be unnecessary to used building materials with either high durability or embodied energy [77].

Buildings are constructed by humans for a variety of purposes. Based on the purpose, different building types have different service life ranges. Canadian Standards Association in CSA S478 has divided different building types into various categories according to their service life [78], and is shown in the Table 2 below:

**Table 2.** Categories of design service lives for buildings [78,79].

| Category | Design Service Life for Category | Examples |
|---|---|---|
| Temporary - | Up to ten years | Non-permanent construction buildings, sale offices, bunkhouses, temporary exhibition buildings |
| Medium Life | 25–49 years | Most industrial buildings, most parking structures |
| Long Life | 50–99 years | Most residential, commercial and office buildingsHealth and education buildings Parking structures below buildings designed for long life category |
| Permanent | Minimum period, 100 years | Monumental or heritage buildings (national museums, art galleries, archives) |

*1.3. Materials Service Life*

The choice of building materials in construction is based on both objective and subjective reasons of both designers and clients. Nevertheless, each material has a specific service life that defines its practical serviceability: this is the amount of time which a material is expected to be useful or serviceable. There are two international standards which define set requirements for "Service life planning" in relation to buildings, components and materials. These are the ISO 15686-1 [76] and ISO 15686-2 [80], which provide a broad design framework necessary in considering various technical, economical, and environmental aspects.

This pre-assessment recommends an approach which is called the "factors method". It uses a factorial approach which draws knowledge relating to the building materials and technology used. It is based largely on an understanding of and transfer of knowledge regarding a material from a previous reference condition to a specific project condition; consequently, the initial reference service life is needed in applying the method when studying building materials and components [81]. Some of the key factors critical to the approximating a product's service life include durability and quality of the base material, as well as design and detailing. Other factors related to workmanship and maintenance regime and levels. Workmanship and design and detailing related factors affect the embodied energy of the materials in buildings due to increase in their maintenance and replacement rates, as well as operational energy due to the thermal performance of the assemblies, where they have been used [82]. The degree of exposure either to human or environmental induced deteriorating effects are also important factors affecting the service life of materials.

There are some other factors which are not included in the "factors method" and can affect the service life of materials. Early demolition of a building due to changing needs of users or due to the economic factors can result in reducing the service life of a materials. Due to such limitations of the factors method, some researchers have asserted that real-life data, as well as an extensive service life database, are needed [36,81].

In line with the current study, there is a relationship between the service life of building materials with recurrent embodied energy. In principle, if the service life of a material is relatively low, and the quantity and number of replacements or repairs that this material or a component made from is higher. Meaning that the magnitude of the embodied energy associated with both manufacturing and installing of this material or component increases. This importance of this consideration is heighted if the production process primarily involves the use of large amounts of fossil fuel-based energy. Consequently, the recurrent demand for energy for this material will have a significant and incremental burden on both the point of origin, and the local environment. Thus, over the whole building life cycle, it may continue to contribute to greenhouse gases emissions into the atmosphere.

## 2. Materials and Methods

In order to determine the effect of variation in the service life of residential buildings and their constituent materials, a detached house in Melbourne was used as case study. A detailed bill of quantities was used to quantify the initial embodied energy and recurrent embodied energy. Average service values for residential buildings and materials were gathered from the literature review, and were used to calculate the recurrent embodied energy, (see step 1 in Figure 1). Average service life of 50 years was used for the case study building for this analysis. After calculating life cycle embodied energy of the house with an average material and building life, a number of variations to service life of building (100 and 150 years) and its constituent materials (minimum and maximum) were made, and life cycle embodied energy was re-calculated, (see step 2–4 in Figure 1).

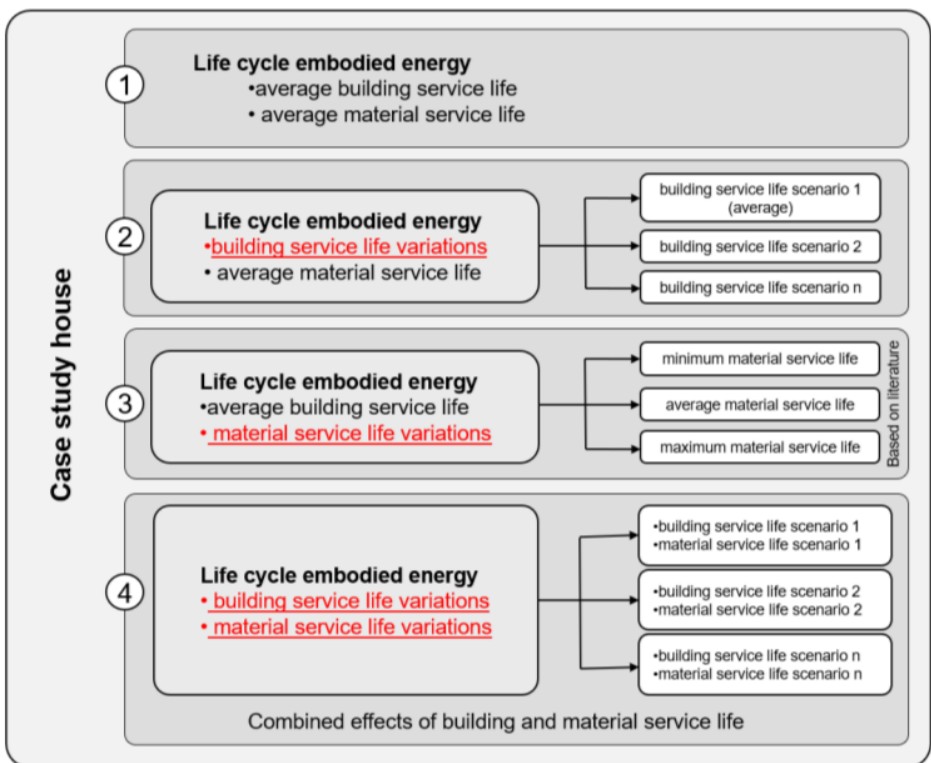

**Figure 1.** Steps taken during the research (adapted from [36]).

### 2.1. Case Study House

The case study house used in the life cycle embodied energy calculations is located in Melbourne. This house has covered area of 292 m$^2$ and has a concrete slab, brick veneer external walls, aluminum windows and terracotta roof tiles, as shown in Figure 2. A detailed bill of quantities was used to quantify the life cycle embodied energy of the house.

### 2.2. Calculating Initial and Recurrent Embodied Energy

Initial embodied energy of the case study house was calculated by using input–output-based hybrid analysis. Delivered quantities of materials used in the construction of the house were multiplied by the hybrid embodied energy coefficient of the respective material, to determine the process-based hybrid embodied energy of the house. Table 3 shows this process for selected materials. To complete the system boundary, the energy embodied in nonmaterial inputs (i.e., the provision of finance, insurance, transport, etc. needed to support the construction process) was calculated, referred to as the remainder of energy inputs, and added to the process-based hybrid embodied energy figure. A detailed description about the use of the input–output-based hybrid analysis to calculate the initial embodied energy of the case study house is available in reference [42].

The recurrent embodied energy of the house was calculated based on the number of times each individual material would likely be replaced during the service life of the house. Criteria for estimating the service life of case study house is described in Section 2.2.1. Material replacements were bases on the service life of each material. Section 2.2.2 describes the method to estimate the service life of materials.

The embodied energy associated with the materials being replaced over the life of the house was calculated as per the initial embodied energy of the house. The delivered material quantities associated with each replacement were multiplied by the respective material embodied energy coefficients. These values included the direct and indirect energy associated with the manufacture of materials. To complete the system boundary, the nonmaterial inputs or remainder associated with materials being replaced, were then calculated as per the initial embodied energy calculation.

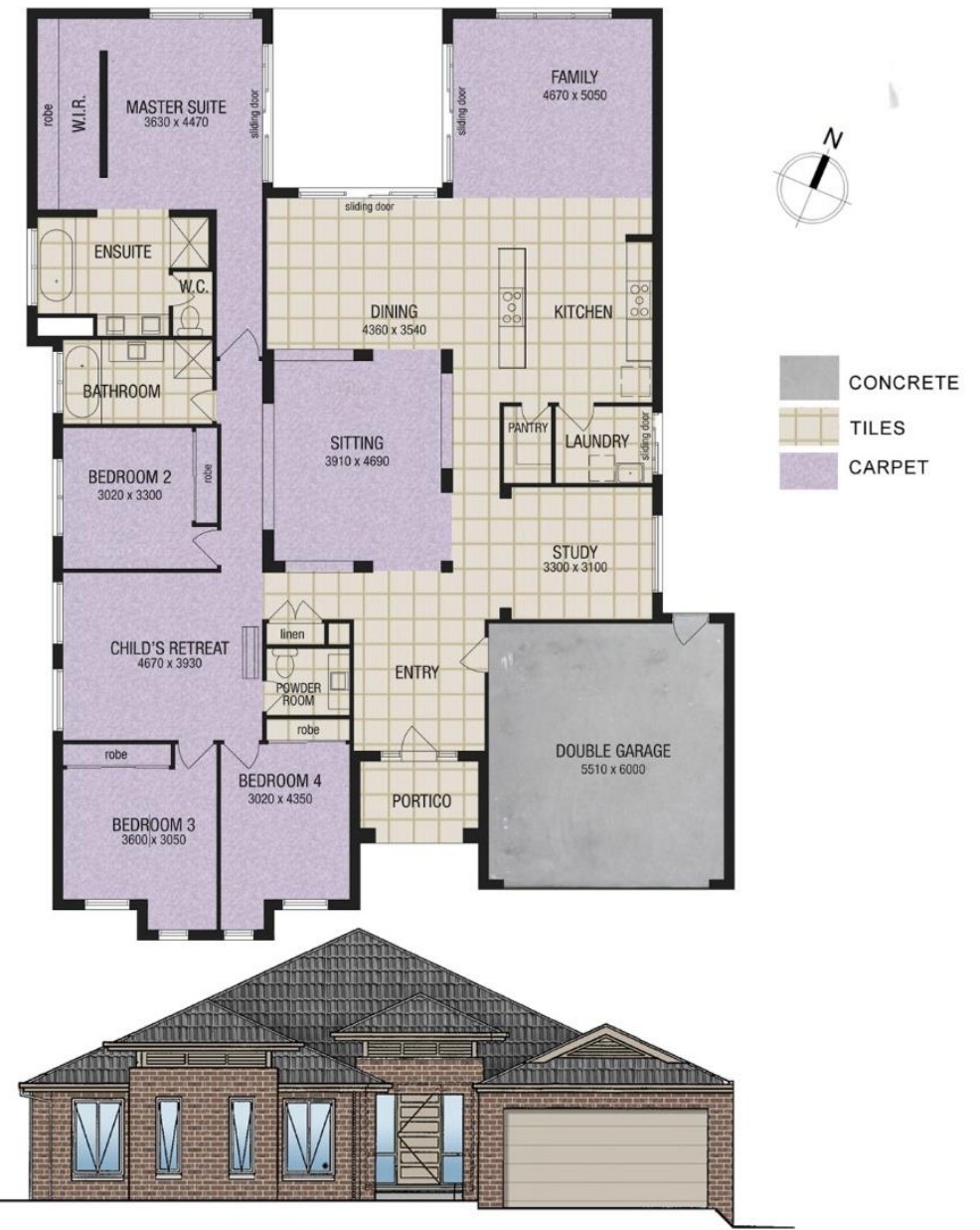

**Figure 2.** Floor plan and elevation of the case study house.

**Table 3.** Calculation of process-based hybrid-embodied energy for the selected materials of case study building.

|  | Quantity | Wastage Factor | Delivered Quantity | Unit | Embodied Energy Coefficient (GJ/Unit) | Embodied Energy (GJ) |
|---|---|---|---|---|---|---|
| Concrete 20 MPa | 52 | 1.15 | 59.54 | m$^3$ | 4.44 | 264.13 |
| Steel | 5.18 | 1.05 | 5.44 | t | 85.46 | 464.61 |
| Timber—Softwood for wall frame | 8.11 | 1.02 | 8.27 | m$^3$ | 10.925 | 90.37 |
| Bricks | 82.89 | 1.05 | 87.03 | m$^2$ | 0.56 | 48.78 |
| concrete roof tile (20 mm) | 378 | 1.10 | 415.61 | m$^2$ | 0.251 | 104.46 |
| carpet—nylon | 240.51 | 1.05 | 252.53 | m$^2$ | 0.683 | 172.54 |
| Paint—water based | 1294 | 1.05 | 1358.70 | m$^2$ | 0.096 | 130.96 |
| Ceramic tiles | 57.1 | 1.05 | 59.96 | m$^2$ | 0.293 | 17.58 |
| Plasterboard 10 mm | 683.15 | 1.05 | 717.31 | m$^2$ | 0.207 | 148.31 |

The energy embodied in each material was then multiplied by the number of replacements for that material over the life of the house, and summed to determine the total recurrent embodied energy associated with the house. The exact number of replacements required for each material was determined by dividing the service life of the house, by the service life of the material, subtracting 1 (representing the material used in initial construction at Year Zero) and rounding up to the nearest whole number (to reflect the fact that materials can only be replaced in whole numbers). A detailed description about the use of the input–output-based hybrid analysis to calculate the recurrent embodied energy of the case study house is available in reference [42].

2.2.1. Estimating Average Building Service Life and Additional Scenarios

Building service life has potential to affect the life cycle embodied energy of a building. The most typical service life found in the literature for residential buildings is 50 years. Therefore, this value was used in the life cycle embodied energy analysis of the case study building.

Variations in building service life may affect the life cycle embodied energy result of a building. Therefore, different building service life scenarios were considered for the calculation of embodied energy. A collation between embodied energy consumption for these building service life scenarios assists in understanding the importance of maximizing building service life in efforts to reduce the building life cycle embodied energy. For this purpose, in addition to the 50 years, life cycle embodied energy was calculated for the building service life of 100 and 150 years.

In order to determine the effect of variations in building service life on the life cycle energy demand associated with the provision of housing over a longer period than the typical life of a house, life cycle embodied energy calculations were conducted for an assessment period of 150 years.

2.2.2. Estimating Material Service Life and Additional Scenarios

This study seeks to investigate the combined effects of material and building service life variations on life cycle embodied energy demand of residential buildings. Therefore, in addition to average material service life values, life cycle embodied energy with the minimum and maximum material service life values was also calculated for building service life of 100 and 150 years.

Material service life figures from the literature were assumed for the analysis (see Table 4). Service life values for the structural materials, such as wall framing, roof timber trusses and concrete slab were assumed similar to the life of building.

**Table 4.** Minimum, average and maximum service life values for selected materials.

| Material | Service Life (Minimum) | Service Life (Average) | Service Life (Maximum) |
|---|---|---|---|
| Concrete roof tiles | 30 | 40 | 60 |
| Bricks | Lifetime | Lifetime | Lifetime |
| Aluminum-framed windows | 10 | 25 | 40 |
| Concrete slab | Lifetime | Lifetime | Lifetime |
| Wall framing | Lifetime | Lifetime | Lifetime |
| Ceramic wall and floor tiles | 20 | 60 | 100 |
| Nylon carpet | 7 | 10 | 20 |
| Paint (interior) | 5 | 10 | 15 |
| Gypsum plasterboard | 20 | 35 | 70 |

## 3. Results and Discussion

In this section, we report the results of the two assessment protocols which were carried out in line with the main and secondary aim of this paper. Sections 3.1–3.4 report the life cycle assessment results of the residential building, highlighting the impact of average,

minimum and maximum service life variations of the building materials and components, as well as variations in the building service life. Section 3.5 is focused on the results of the Sensitivity analysis which was conducted to report the impact of varying the assessment period in line with changes in both material and building service life values simultaneously.

### 3.1. Base Case Results (with Average Material and Building Service Life)

The embodied energy associated with the initial construction of the case study house was found to be 3891 GJ (13.4 GJ/m$^2$). It includes the energy embodied in the manufacturing of materials, transportation, initial construction of the house and supporting services. Recurrent embodied energy associated with replacement of materials with average material service life values over an average building service life of 50 years and was found to be 2789.5 GJ (9.55 GJ/m$^2$).

Life cycle embodied energy of the case study house was calculated by adding the initial and recurrent embodied energy. Life cycle embodied energy of the house with average material service life values for a period of 50 years was found to be 6680.5 GJ (22.9 GJ/m$^2$). Initial and recurrent embodied energy were found to constitute 58% and 42% of the life cycle embodied energy, as shown in Figure 3.

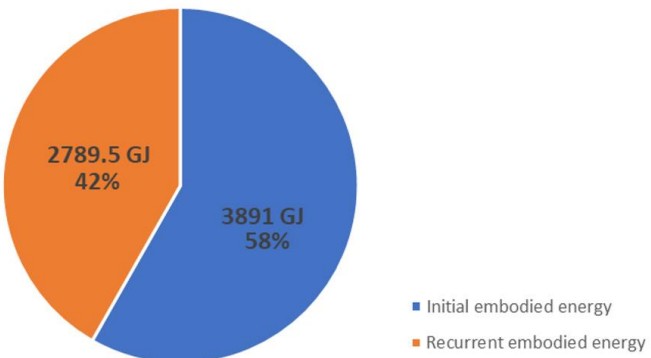

**Figure 3.** Proportion of the initial and recurrent embodied energy.

The findings of the IOBHA for the current study shows the importance of embodied energy assessment, particularly in line with life cycle assessments, as well as the huge emphasis on operational energy. Though are no rigid international standards yet, which define all factors essential to LCEE studies; however, comparing results from other studies is helpful. Comparatively speaking, we reviewed a few studies and found that the floor area, geographical location and assessment method, among other factors mentioned elsewhere [37], may impact these results. For example, another study with predominant use of concrete, a building floor area of 532 m$^2$, the LCEE was 13473 GJ (25.5 GJ/[37], while LCE proportions were 36% (LCEE) and 64% (LOPE) [37]. In this example, the high LOPE was due to extensive air conditioning in the arid long summer months and high LCEE due to extensive use of high-energy intensive materials. Other authors report that LCEE/m$^2$ values of 11.62 [48] and 3.45 [73].

In general, LCEE studies cannot be generalized without contextualisation and proper material inventory. The LCEE/m$^2$ for the current study gives a relatively higher figure that the other studies due to the IOBHA method used, which provides a more comprehensive assessment tool. Another reason may include the extensive material and component inventory used in this study.

### 3.2. Effect of Material Service Life Variations

The recurrent embodied energy associated with the replacement of materials for the detached house over a period of 50 years, based on minimum and maximum material service life figures, were found to be 5335 GJ (18 GJ/m$^2$) and 1756 GJ (6 GJ/m$^2$), respectively, as shown in Figure 4.

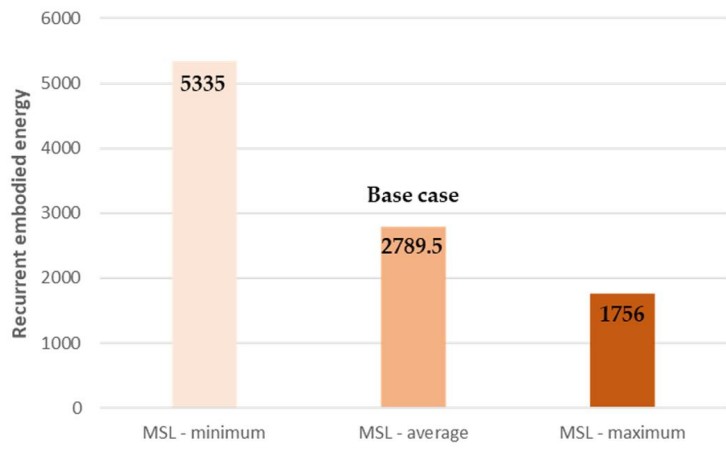

**Figure 4.** Recurrent embodied energy of the case study house with the minimum, average and maximum material service life over 50 years.

Life cycle embodied energy of the detached house for minimum, average and maximum material service life scenarios was found to be 9226 GJ (31.7 GJ/m$^2$), 6680.5 GJ (22.9 GJ/m$^2$) and 5647 GJ (19.4 GJ/m$^2$), as shown in Figure 5 below. The life cycle embodied energy over a period of 50 years, based on minimum material service life values is almost 140% more than the life cycle embodied energy of the house with average material service life values or about 38% more than the life cycle embodied energy of the house with average material service life values. The life cycle embodied energy of the house over the same period, based on maximum material service life values is about 85% of the life cycle embodied energy of the house with average material service life values. Life cycle embodied energy was found to have decreased by 39%, when the minimum material service life scenario was changed to the maximum material service life scenario. This shows that when materials are poorly maintained and/or require greater frequency of replacement, life cycle embodied energy of a building may increase significantly. It also results in recurrent embodied energy representing a higher proportion of life cycle embodied energy of a building.

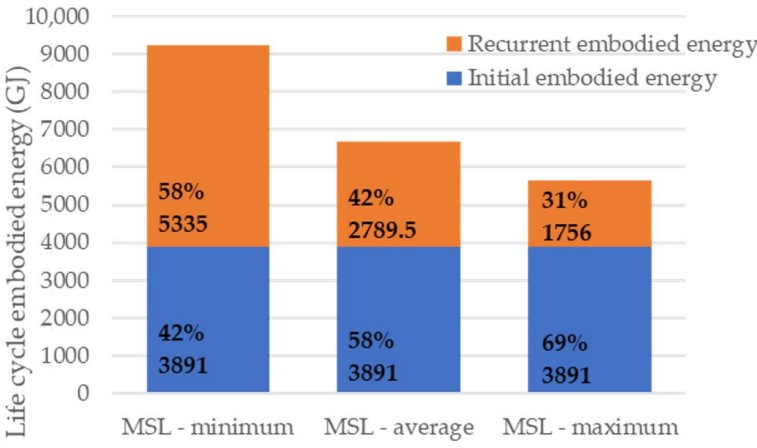

**Figure 5.** Life cycle embodied energy of the case study house with the minimum, average and maximum material service life over 50 years.

### 3.3. *Effect of Building Service Life Variations*

In order to determine the effect of variations in building service life on the life cycle embodied energy demand associated with the provision of housing over a longer period than the typical life of a house in Melbourne, life cycle embodied energy of the house with building service life of 50, 100 and 150 years was calculated for an assessment period of 150 years.

### 3.3.1. Initial Embodied Energy

Initial embodied energy is not affected by increasing the service life of building. Over an assessment period of 150 years, for a building service life of 50, 100 and 150 years, the cumulative initial embodied energy was 11,673 GJ, 7782 GJ and 3891 GJ (equal to the initial embodied energy of the house), respectively. This means that for a building service life of 50 years, the house is built a total of three times over 150 years and the total cumulative initial embodied energy over this period is three times the initial embodied energy of the house (ignoring possible variations in this figure over time due to changes in materials used, their production process and other factors).

### 3.3.2. Recurrent Embodied Energy

Recurrent embodied energy increased with an increase in service life of buildings, as more materials are being replaced over the longer service life of buildings. Recurrent embodied energy was found to increase from 2789.5 GJ at a building service life of 50 years to 6300.6 GJ and 10,248 GJ for the building service life of 100 and 150 years, respectively.

For an assessment period of 150 years, recurrent embodied energy for the building service life of 50, 100 and 150 years was also calculated. This means, for a building service life of 50 years, the recurrent embodied energy was calculated for 50 years and multiplied by the number of whole building service life periods within the 150 years ($150/BSL$) $\times$ $REE_{BSL}$). In the case, where the number of whole building service life periods was not a whole number, recurrent embodied energy for the remaining period was also added. Over an assessment period of 150 years, for a building service life of 50, 100 and 150 years, the recurrent embodied energy was 8368.5 GJ, 9090.1 GJ and 10,248 GJ, respectively.

### 3.3.3. Life Cycle Embodied Energy

Life cycle embodied energy of the house with an average material service life was found to be 6680.5 GJ (457.6 MJ/m$^2$/year), 10,080 GJ (346 MJ/m$^2$/year) and 14,027 GJ (321 MJ/m$^2$/year) for the building service life of 50, 100 and 150 years, respectively, as shown in Figure 6. Although the cumulative embodied energy is on the rise, the associated embodied energy, calculated on an annual basis is actually on the decline. On an annual basis, this shows a 24% and 29% decrease in the life cycle embodied energy demand, when compared to the building service life of 50 years. This shows the advantage of prolonging a building's service life in order to optimize its life cycle embodied energy demand.

For an assessment period of 150 years, life cycle embodied energy of the house with an average material service life was found to be 16,222 GJ (370 MJ/m$^2$/year) and 13,296 GJ (304 MJ/m$^2$/year) for the building service life of 100 and 150 years, respectively (see Figure 7). On an annual basis, this shows a 13% and 29% decrease in the life cycle embodied energy demand, when compared to the building service life of 50 years (18,629 GJ or 425 MJ/m$^2$/year). This again demonstrates the advantage of prolonging the service life of buildings in order to optimize their life cycle embodied energy demand.

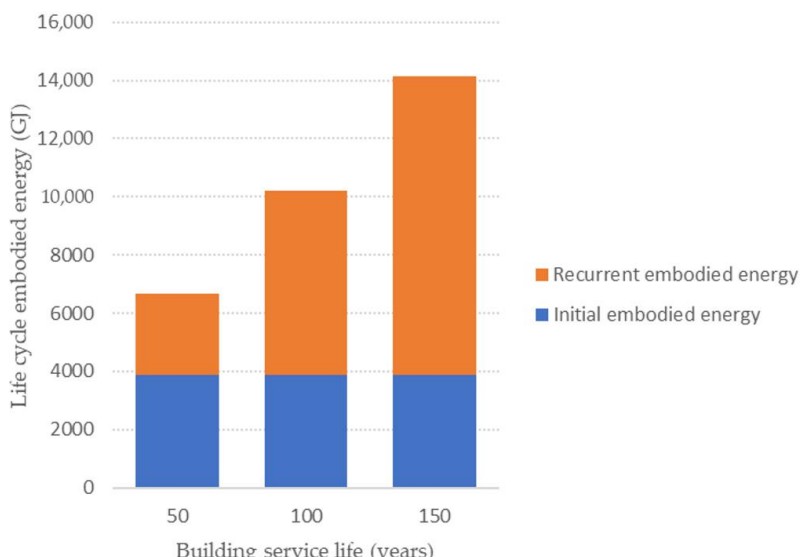

**Figure 6.** Life cycle embodied energy of the case study house with the average material service life for building service lives of 50, 100 and 150 years.

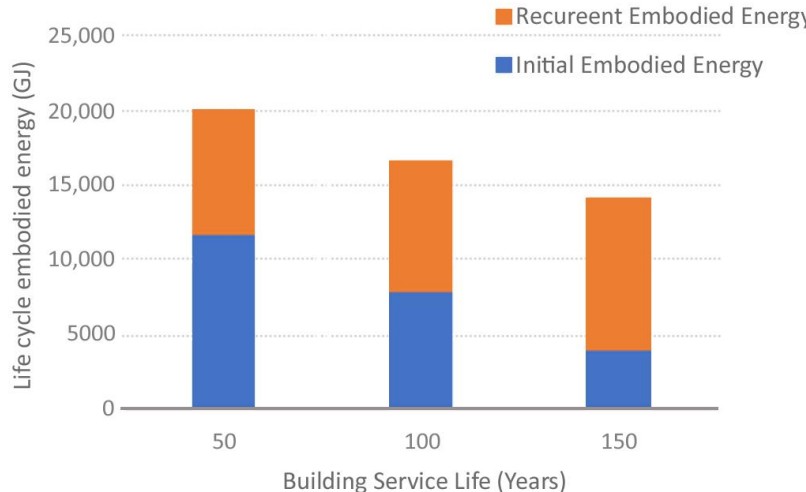

**Figure 7.** Life cycle embodied energy of the case study house with the average material service life for building service lives of 50, 100 and 150 years over an assessment period of 150 years.

### 3.4. Combined Effect of Building and Material Service Life

Firstly, the life cycle embodied energy of the house with the minimum and maximum material service life was recalculated, the comparative magnitudes are significantly differences. For minimum values, the life cycle embodied energy was 9226 GJ (632 MJ/m$^2$/year) 15,376 GJ (526.6 MJ/m$^2$/year) and 21,943 GJ (501 MJ/m$^2$/year) for the building service life of 50, 100 and 150 years, respectively. For maximum values, the life cycle embodied energy was 5647 GJ (386.8 MJ/m$^2$/year) 8056 GJ (275.9 MJ/m$^2$/year) and 10,884.5 GJ (248.5 MJ/m$^2$/year). These results are displayed in Figure 8 below.

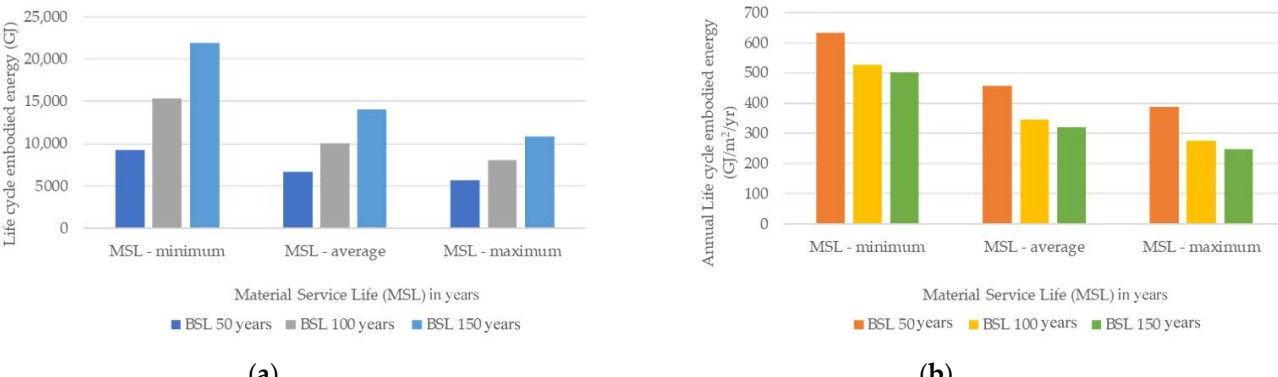

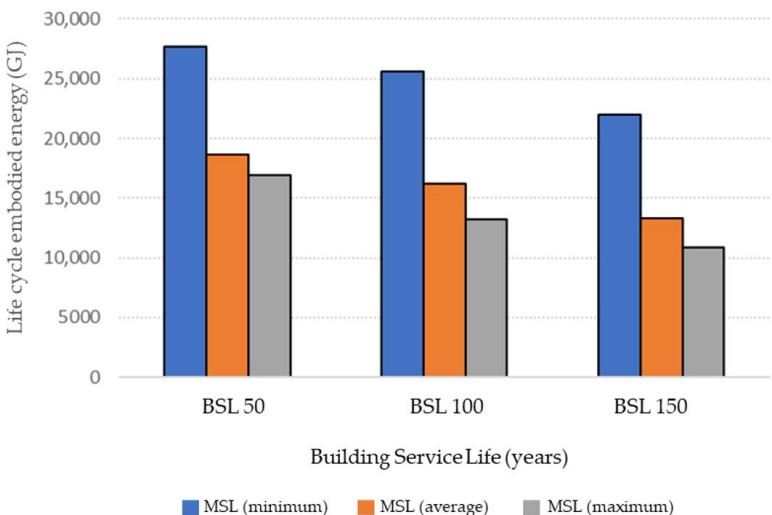

**Figure 8.** (**a**) Life cycle embodied energy (in GJ). (**b**) Annual life cycle embodied energy (in $GJ/m^2/year$) of the case study house with material service life variations for building service lives of 50, 100 and 150 years.

Life cycle embodied energy with an average material service life over an assessment period of 150 years was found to be 18,629 GJ, 16,222 GJ and 13,296 GJ for building service life of 50, 100 and 150 years, respectively. With the minimum material service scenarios, life cycle embodied energy over the same assessment period increased to 27,679 GJ, 25,625 GJ and 21,944 GJ for building service life projections of 50, 100 and 150 years, respectively. Life cycle embodied energy with the maximum material service life over the same assessment period decreased to 16,943 GJ, 13,224 GJ and 10,885 GJ for building service life projections of 50, 100 and 150 years, respectively (see Figure 9). This shows that life cycle embodied energy decreases with an increase in the service life of materials. This figure also shows that an increase in building service life results in a decrease in life cycle embodied energy demand over the assessment period of 150 years. The extent of the decrease in life cycle embodied energy demand was determined by analyzing the variation between the highest and lowest embodied energy results.

**Figure 9.** Life cycle embodied energy of the case study house with the minimum, average and maximum material service lives for 50, 100 and 150 years over an assessment period of 150 years.

Life cycle embodied energy with the minimum material service life for a building service life of 50 years was found to be the highest (at 27,679 GJ or 95 $GJ/m^2$). The main reason behind this high value of embodied energy for the building service life of 50 years was the need for two complete replacements of the house during the 150 years assessment period. This shows the importance of prolonging the service of buildings. Despite two whole building replacements, life cycle embodied energy with the average (18,628 GJ or

64 GJ/m$^2$) and maximum material service life (16,943 GJ or 58 GJ/m$^2$) for a building service life of 50 years was significantly lower than the life cycle embodied energy with the minimum material service life. This shows the importance of prolonging the service life of materials. Life cycle embodied energy with the maximum material service life for a building service life of 150 years was found to be the lowest (at 10,855 GJ or 37 GJ/m$^2$). Variation between the highest result (27,679 GJ for minimum material service life for a building service life of 50 years) and lowest result (10,855 GJ for maximum material service life for a building service life of 150 years) was found to be 61% (2.55 times). This shows that extending the service life of buildings and their constituent materials results in a considerable reduction in their life cycle embodied energy demand. Usually, an increase in building and material service life has no, or limited adverse effect on the operational energy requirements. Therefore, a reduction in life cycle energy demand of buildings and associated environmental effects is possible by careful consideration of building and material service life planning at design stage.

### 3.5. Sensitivity Analysis

In the literature, the use of a sensitivity analysis facilitates in decision making and is used as a means of exploring present or future variations which may impact the results of a study [42,83,84]. In previous studies, the use of a sensitivity analysis showed that embodied energy analysis may have an error of up to 42% [42]. This section elaborates the reason, procedure and results of an extended sensitivity analysis which was carried out.

The scope of the current analysis was to explore the possible impact of variations changes in material service life values. Minimum, average and maximum MSL values were used to calculate the embodied energy associated with the selected case study building. The values used were extracted from the literature which was referenced in Section 1.3, and elaborated elsewhere [36]. While average MSL values represent the most common values found in the literature, the minimum was the least value and maximum was the highest value found. However, some factors may lead to a change in the MSL such as user behavior, building location, weathering and durability, level of craftmanship and time-induced corrosion [85–87]. An individual or combined impact of these factors may consequently cause a drop below the minimum or a rise above the maximum material service life values of individual products. Therefore, we explored alternate scenarios which a 20% variation in relation to the minimum, average and maximum material service life values for each building material used. The objective was to observe the impact and further explore the benefits, of increasing the service life, as well as the disadvantages of reducing the material service values of materials. The variations used in this analysis ranged from a 20% decrease in the minimum and average MSL values, as well as a 20% increase in the average and maximum values (Figure 10).

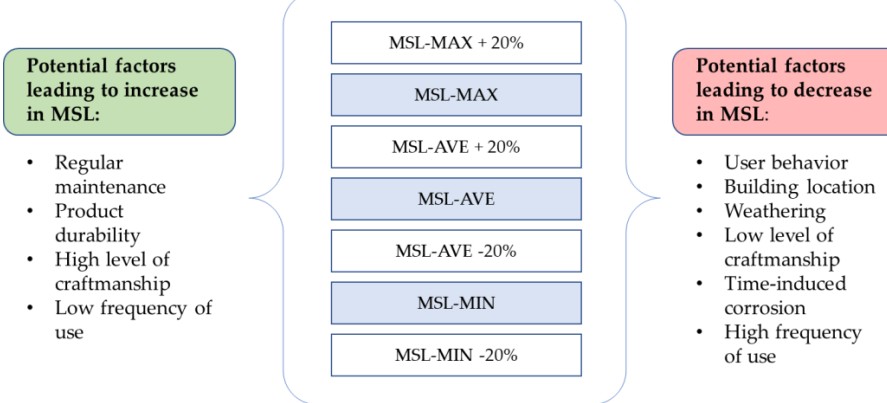

**Figure 10.** MSL variations and potential factors.

Using the same input–output hybrid analysis that was applied to calculate the results presented in Sections 3.1 and 3.3, the initial, recurrent and life cycle embodied energies associated with all alternative variations of the material service life were calculated. Figure 11 is an annotated table which shows the results of the analysis which is explained in the light of the life cycle embodied energy and the assessment periods of 50 and 150 years in Sections 3.5.1 and 3.5.2.

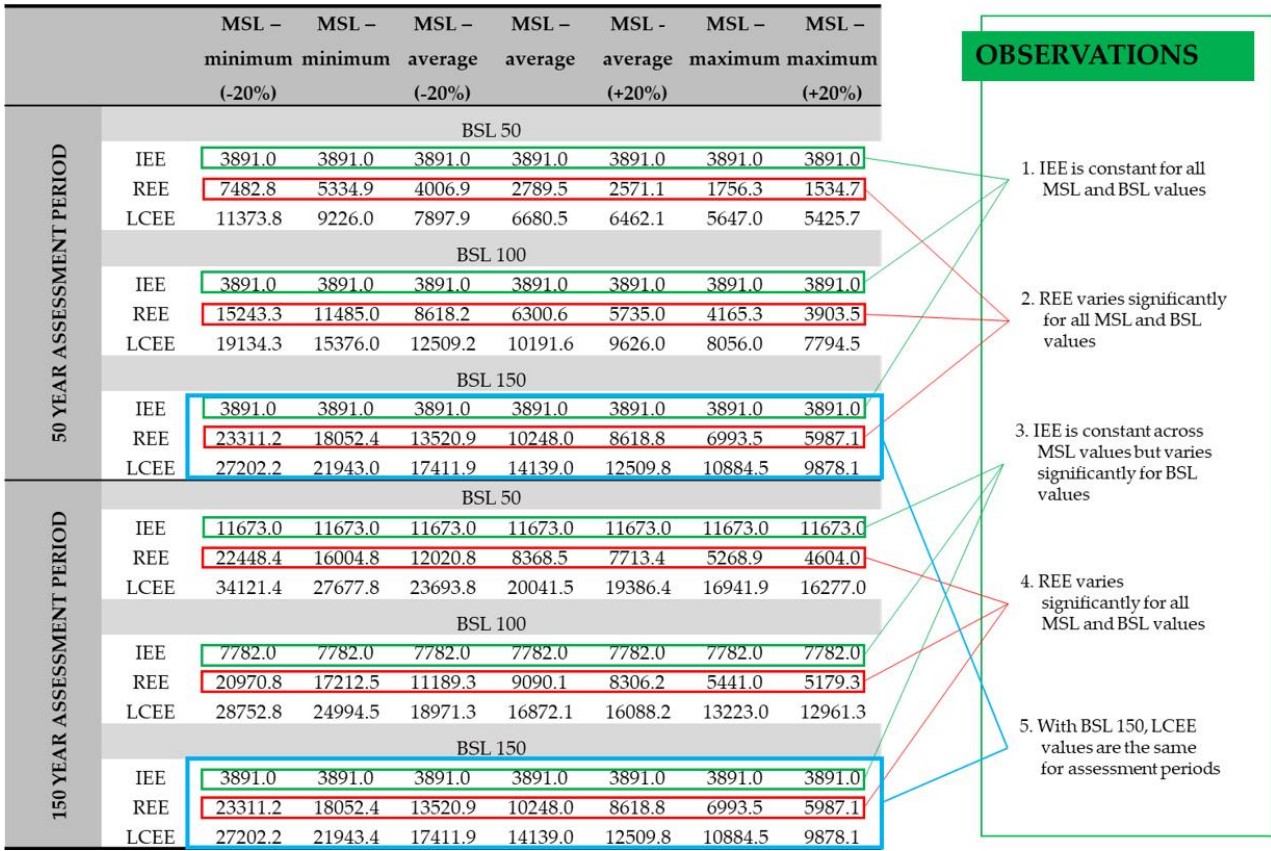

**Figure 11.** Annotated sensitivity analysis for MSL, BSL and assessment period data.

The figure shows five key points to note; firstly, for the 50-year assessment period, the initial embodied energy associated with all MSL and BSL variations is constant. This is because the building will only be constructed once during the assessment period; that is, the assessment period is either equal to or greater than the building service life. Secondly, the recurrent embodied energy varies significantly for all MSL and BSL values. As reported in Section 3.3, the recurring need for repair and maintenance is constant regardless of the MSL or BSL values. Although the rate of increase differs, the increase is certain. The percentage variations of these increments are discussed further in Sections 3.5.1 and 3.5.2. Thirdly, for the 150 years assessment period, while the initial embodied energy remained constant across all MSL values, it varies significantly for BSL values. This is likely because for a longer assessment period the initial energy associated with construction does not increase regardless of the MSL values. However, when the BSL values increases, the number of re-builds reduces; in this case, for the same period, the building would be constructed one time (BSL 50), two times (BSL 100) and three times (BSL 150). This implies that for each reconstruction, the initial embodied energy increases by a factor X which is equal to the initial embodied energy when the building was first constructed. This leads to the fourth point: the REE varies significantly for all MSL and BSL values. This is similar to the findings under the 50-year assessment period. Recurrent embodied energy. Finally, when the assessment period is changed from 50 to 150 years, for BSL 150 only, the IEE, REE

and LCEE remain constant. However, the following sections give more insight into the sensitivity analysis results with respect to the assessment periods.

### 3.5.1. Life Cycle Embodied Energy Variations Based on the 50-Year Assessment Period

For building service life values of 50, 100 and 150 years over a 50–year assessment period, Figure 12 shows the variation in the life cycle embodied energy the case study villa. Generally, for each material service life variation, the LCEE increases as BSL increases. Starting with the MSL-minimum (−20%), the average percentage increase was about 55%. However, for the other MSL values, this percentage declines as follows: 55% for MSL minimum, 49% for MSL-average (−20%), 39% for MSL average, 45% for MSL average (+20%), 39% for MSL maximum, and 38% for MSL maximum (+20%).

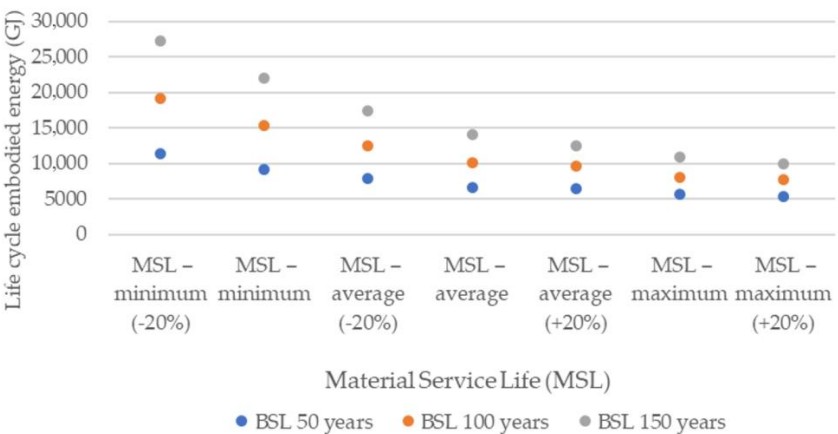

**Figure 12.** Life cycle embodied energy impact of BSL and MSL variations (50-year assessment period).

The figure shows that reviewing the MSL alternatives for BSL 50 the life cycle embodied energy varies from about 11,373 GJ to 5456 GJ. This shows an 208% variation between the least and greatest material service life values. For BSL 100 the life cycle embodied energy varies from 19,134 GJ to 7795 GJ and shows an 246% variation between the least and greatest material service life values. Finally, for BSL 150 the life cycle embodied energy varies from 27,202 to 9878 GJ which represented a 275% variation between the least and greatest material service life values.

### 3.5.2. Life Cycle Embodied Energy Variations Based on the 150-Year Assessment Period

For building service life values of 50, 100 and 150 years over a 150–year assessment period Figure 13 shows the variation in the life cycle embodied energy the case study villa. Similar to Section 3.5.1, this figure also shows the material service life variations. However, in this case, for each material service life variation, the LCEE decreases as BSL increases. For the MSL-minimum (−20%), the percentage drop is about 10.6%. However, for the other MSL values, this percentage increases as follows: 11% for MSL minimum, 14% for MSL-average (−20%), 16% for MSL average, 19.6% for MSL average (+20%), 19.8% for MSL maximum, and 22% for MSL maximum (+20%). This shows that as the MSL increases the sensitivity of the life cycle embodied energy values reduces. This also shows that as the MSL increases, the percentage difference in the life cycle embodied energy becomes more and the sensitivity of the life cycle embodied energy variation increases.

The figure also shows the MSL variations with −20% decrease in minimum MSL and +20% increase in maximum MSL for BSL of 50 years; the life cycle embodied energy varies from 34,121 GJ to 16,277 GJ. This shows an 210% variation between the least and greatest material service life values (add here difference in MSL minimum and maximum). For BSL 100 the life cycle embodied energy varies from 28,753 GJ to 12,961 GJ and shows an 222% variation between the least and greatest material service life values. Finally, for BSL of 150 years, the life cycle embodied energy varies from 27,202 GJ to 9878 GJ, which

represented a 275% variation between the least and greatest material service life values. This confirms that a decrease or increase in the service life of buildings and their constituent materials due to any of the factors discussed in Sections 1.2 and 1.3 can result in significant variation in life cycle embodied energy of a building. Advantage of prolonging the service life of buildings and their material reinforces the importance of building and material service life consideration at any life cycle stage of a building, in particularly at design stage in order to reduce the life cycle embodied energy in built environment.

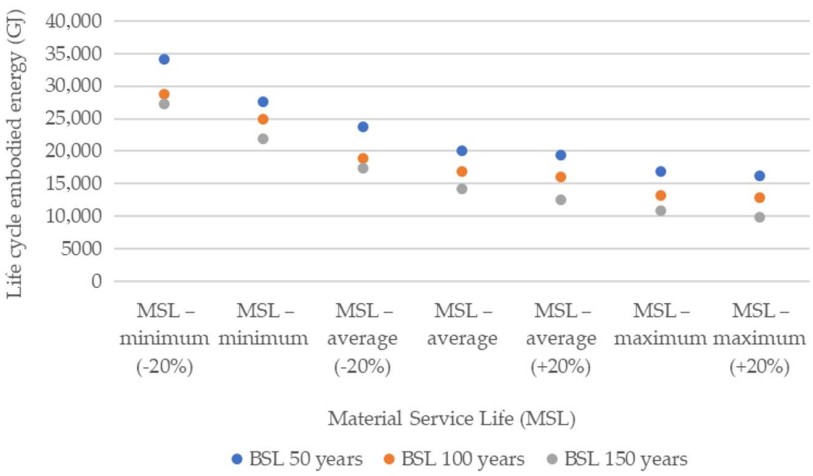

**Figure 13.** Life cycle embodied energy impact of BSL and MSL variations (150-year assessment period).

## 4. Conclusions

The aim of this study was to determine what combined effect the variation in the service life of materials and buildings has on the life cycle embodied energy demand associated with a residential building. A single-story house was used as a case study for this analysis. The initial and recurring embodied energy of the house for a building service life of 50 years were calculated using a comprehensive hybrid assessment approach, with material service life values based on average figures obtained from the literature. These service life values along with the minimum and maximum material service life values were recalculated over an assessment period of 150 years. This process was repeated for the building service life of 100 and 150 years.

When comparing the life cycle embodied energy results using maximum material service life values with the minimum material service life values, a 39% reduction in life cycle embodied energy was found. A 29% reduction in life cycle embodied energy demand was found when the building service life was extended from 50 years to 150 years over the same assessment period. Life cycle embodied energy results over an assessment period of 150 years shows that residential buildings with short life spans are particularly bad for environment due to the investment of embodied energy multiple times over a longer span of time to fulfil the housing needs. Combined effect of maximizing the material and building service life resulted in even higher savings in life cycle embodied energy with a reduction of 61% life cycle embodied energy demand.

The sensitivity analysis results show the relationship between the magnitude of the life cycle embodied energy with material service life, building service life and assessment period variations. Generally, when material service life is at a minimum, and building service life is at a maximum, the life cycle embodied energy is at a maximum (27,202 GJ) during the 50-year assessment period. However, when the material service life is at a maximum and building service life is at a minimum, the life cycle embodied energy is at a minimum (9878 GJ). The sensitivity analysis showed the impact of material service life variations ranges from 20% to 55% for the 50-year assessment period, and from 10.6% to 22% for the 150-year assessment period.



This study and the sensitivity analysis have shown that a variation in the service life of materials and buildings can affect the life cycle embodied energy significantly. However, their combined effect has resulted in a significantly higher degree of variation in life cycle embodied energy demand, showing the importance of choosing the materials with their service life in conjunction with the service life of the buildings. This demonstrates that in an attempt to reduce the life cycle embodied energy demand of buildings to minimize the associated environmental impacts, it is important that the service life of buildings and their constituent materials be taken into consideration.

**Funding:** This work was funded by a United Arab Emirates University Start-Up Grant (G00003524).

**Institutional Review Board Statement:** Not applicable.

**Informed Consent Statement:** Not applicable.

**Data Availability Statement:** Not applicable.

**Conflicts of Interest:** The author declares no conflict of interest.

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
