# Peer review of "Reducing Life Cycle Embodied Energy of Residential Buildings: Importance of Building and Material Service Life"

_buildings, doi:10.3390/buildings12111821_

Round 1

Reviewer 1 Report

The article “Reducing life cycle embodied energy of residential building and material service life” deals with an interesting and novel topic which is the combined assessment of the building service life and materials service life. However, the following major aspects should be addressed before acceptance:

- The complete article results were no introduced, the sensibility analysis for example was no included in introduction.

- Use superscript in all corresponding units. For example in GJ/m2

- The results and conclusions need to be compared with existing literature or just “yardsticks”. For example, results must be compared to energy data from known cities or person consumption, among others.

- Figure 2 is blurry, improve this. Also, the orientation from top view with frontal view should be revised.

- Although the methodology used for the building service life and materials service life assessments were included, not all information was presented to readers. Using the traditional four stages is highly recommended. This means goal and scope definition, inventory analysis, impact assessment, analysis of results.

- In line 397 verify this data and question mark: (or 39?)

Reviewer 2 Report

This paper studies relationship between energy consumption of the residential building life cycle and service life of the building materials. The main problems in this paper are as follows:

1. What are the innovative points of this paper, which should be clearly presented in the abstract and introduction.

2. The clarity of the pictures in this article needs to be improved.

3. The references cited in this paper are old, it is suggested to quote more references in the past five years.

4. The format of this article still has some problems and needs to be modified.

5. Overall the author of this paper has done a good job and recommends that it is accepted after minor revision.

Reviewer 3 Report

The manuscript aims to investigate the life  cycle energy demand of residential buildings as a consequence of its service life, as well  as the service life of its construction materials.

The content of the paper corresponds to the topic stated in the title.

The article has been correctly divided into four chapters:

  1. Introduction
  2. Methodology
  3. Results and discussion
  4. Conclusions

The work contains 13 figures and 3 tables.

The aim of the study is to investigate the combined effect of building and material service life on life cycle embodied energy requirements of residential buildings.

The aim of this study has been achieved.

The sources contain only 27 references, of which 4 are self-citations.

I recommend the article for publishing after taking into account the following remarks:

1.       Chapter 2. “Methodology” should be named “Materials and Methods”

2.       All diagrams should include units on the x-axis and y-axis. This should be added. Foe example figure 8.

3.       Sources are missing in figure descriptions. This should be added.

4.       Chapter 2 states that “A detailed bill of quantities was used to quantify the initial embodied energy and recurrent embodied energy” and “Delivered quantities of materials used in the construction of the house were multiplied by the hybrid embodied energy coefficient of the respective material, to determine the process-based hybrid embodied energy of the house.”.

The bill of quantities is a large file and would be impossible to include in an article. However, could you add a table where you show, for example, 10 selected items from the bill of quantities and their corresponding hybrid embodied energy coefficient and the result of multiplication.

5.       On page 3 is Table 1, on page 5 is Table 2 and an on page 8 is Table 1. Please edit on page 8 to "Table 3".

6.       Page 8 “Material service life figures from the literature were assumed for the analysis (see Table 2).”. Please edit to “(see Table 3)”.

7.       In both the text and figures, numbers should be separated by a dot instead of a comma, for example, on Figure 8a and  figure 9 it is "5,000” and should be "5.00"

Round 2

Reviewer 1 Report

Dear author, 

Dear author,

Many thanks for considering most review comments, the article "Reducing life cycle embodied energy of residential buildings: importance of building and material service life", in my opinion, could be publish in Buildings.

Best regards,